# Structure–function studies of ultrahigh molecular weight isoprenes provide key insights into their biosynthesis

Hiroyuki Kajiura [1,2,3,7], Takuya Yoshizawa[3,7], Yuji Tokumoto [2,5], Nobuaki Suzuki[2], Shinya Takeno[2], Kanokwan Jumtee Takeno[2], Takuya Yamashita[3], Shun-ichi Tanaka[3], Yoshinobu Kaneko[4], Kazuhito Fujiyama[1], Hiroyoshi Matsumura [3 ✉] & Yoshihisa Nakazawa[2,6 ✉]

Some plant *trans*-1,4-prenyltransferases (TPTs) produce ultrahigh molecular weight *trans*-1,4-polyisoprene (TPI) with a molecular weight of over 1.0 million. Although plant-derived TPI has been utilized in various industries, its biosynthesis and physiological function(s) are unclear. Here, we identified three novel *Eucommia ulmoides* TPT isoforms—EuTPT1, 3, and 5, which synthesized TPI in vitro without other components. Crystal structure analysis of EuTPT3 revealed a dimeric architecture with a central hydrophobic tunnel. Mutation of Cys94 and Ala95 on the central hydrophobic tunnel no longer synthesizd TPI, indicating that Cys94 and Ala95 were essential for forming the dimeric architecture of ultralong-chain TPTs and TPI biosynthesis. A spatiotemporal analysis of the physiological function of TPI in *E. ulmoides* suggested that it is involved in seed development and maturation. Thus, our analysis provides functional and mechanistic insights into TPI biosynthesis and uncovers biological roles of TPI in plants.

[1] International Center for Biotechnology, Osaka University, 2-1 Yamada-oka, Suita, Osaka 565-0871, Japan. [2] Technical Research Institute, Hitachi Zosen Corporation, 2-2-11 Funamachi, Taisyo, Osaka 551-0022, Japan. [3] Department of Biotechnology, College of Life Sciences, Ritsumeikan University, 1-1-1 Noji-higashi, Kusatsu, Shiga 525-8577, Japan. [4] Yeast Genetic Resources Lab, Graduate School of Engineering, Osaka University, 2-1 Yamada-oka, Suita, Osaka 565-0871, Japan. [5] Present address: Department of Evolutionary Biology and Environmental Studies, University of Zurich, Winterthurerstrasse 190, 8057 Zurich, Switzerland. [6] Present address: Faculty of Bioscience and Bioindustry, Tokushima University, 2-1 Minami-josanjima, Tokushima 770-8513, Japan. [7] These authors contributed equally: Hiroyuki Kajiura, Takuya Yoshizawa. ✉email: h-matsu@fc.ritsumei.ac.jp; nakazawa@tokushima-u.ac.jp

Prenyltransferases, also known as isoprenyl diphosphate synthases, catalyze the sequential condensation of a basic five-carbon building block ($C_5$), the isopentenyl diphosphate (IPP), to synthesize prenyl precursors of isoprenoid metabolites and isoprene polymers, collectively called polyisoprene. Ultrahigh molecular weight cis-1,4-polyisoprene and trans-1,4-polyisoprene (TPI), which are the principal components of natural rubber and gutta-percha, respectively, are also isoprenoid metabolites that are assumed to be synthesized by specialized prenyltransferases. These prenyltransferases are divided into two major groups, the cis-1,4-prenyltransferases (CPTs) and trans-1,4-prenyltransferases (TPTs), based on the stereochemistry of the double bond formed during IPP condensation. Although both types of prenyltransferases require an allyl substrate and IPP during the common condensation step, they do not share consensus and conserved sequences, indicating that TPTs and CPTs originated from evolutionarily distinct proteins[1–3]. CPTs form a membrane-associated complex with accessory proteins to generate ultrahigh molecular weight cis-1,4-polyisoprene, whereas the molecular mechanism of TPTs and the physiological functions of TPI remain largely unclear.

Farnesyl diphosphate synthase (FPS) represents a well-characterized TPT prototype. It is homodimeric, and each monomer has seven highly conserved TPT-specific domains and two Asp-rich motifs (DDxxD), which are essential for the interaction with IPP and the allyl substrate[1,3]. The FPS dimer consists of the α-helical isoprenoid fold with a large central cavity[4,5]. Previous mutagenesis studies demonstrated that the side chains of the bulky amino acids Phe112 and Phe113 (in avian numbering) at the fourth and fifth position upstream of the first DDxxD motif are critically involved in chain length determination of the polymeric product[6,7]. Typically, FPS produces TPI with a chain length range of $C_{10–40}$, but enzymes with mutations in Tyr88 and Phe89 generate TPI with a chain length of up to $C_{50}$[7]. However, some plant species, including Eucommia ulmoides, Palaquium gutta, Manilkara bidentata, and Manilkara zapota, produce ultrahigh molecular weight TPI with a chain length of $C_{100}$ or much more ($C_{50,000}$, estimated from the maximum molecular weight)[8–10], which indicates the biosynthetic activity of unique TPTs. Especially E. ulmoides has become highly accessible for further research due to the availability of the dataset derived from an expressed sequence tag (EST) library[11]. Furthermore, a TPT candidate for the synthesis of ultrahigh molecular weight TPI in E. ulmoides was proposed based on the recently sequenced genome of this tree species[12].

In this study, we searched an E. ulmoides EST library and identified three potential TPTs, thereafter referred to as EuTPT1, 3, and 5. We aimed to investigate the involvement of these enzymes in TPI production and elucidate the potential molecular mechanism of this biosynthetic pathway. The in vitro analysis of the EuTPTs included a crystallography study of EuTPT3 to further unravel the structural requirements for the synthesis of ultrahigh molecular weight TPI and the formation of the enzyme core of ultralong-chain TPTs. An accompanying in vivo analysis of EuTPT5 had the objective to elucidate the physiological function of this enzyme and the involvement of its product, TPI, in fruit development and maturation in E. ulmoides.

## Results

### The Eucommia ulmoides genome encodes three non-FPS family prenyltransferases.

We previously searched the database of our E. ulmoides EST library[11] using FPS amino acid sequences from Arabidopsis thaliana (AtFPS1 and 2) and Saccharomyces cerevisiae (YJL167W, ScFPS) as queries. This search identified five potential EuTPTs, EuTPT1 to EuTPT5[11] (Fig. 1a). All EuTPTs have seven highly conserved TPT-specific domains containing DDxxD motifs, but EuTPT1, 3, and 5 shared six additional amino acid insertions that distinguished them from AtFPSs, ScFPS, EuTPT2, and EuTPT4 (Supplementary Fig. S1). Moreover, amino acid residues of chain length determination in EuTPT1, 3, and 5 were Cys-Ala, unlike the bulky amino acids Tyr/Phe-Phe found in the other FPS sequences and in EuTPT2 and 4[11,12].

To characterize the function of the potential EuTPTs, a complementation analysis was performed as a first screening using S. cerevisiae Δfps/FPS heterozygous diploid mutant and the five corresponding EuTPT cDNAs. The ScFPS gene is an essential gene and the null mutation (Δfps) shows inviable phenotype[13]. If any of the EuTPT could rescue the Δfps lethality, the encoded EuTPT was identified as an FPS and ruled out as an ultralong-chain TPT candidate. In the tetrad analysis, ScFPS, EuTPT2, and EuTPT4 rescued Δfps lethality. It should be noted that we previously characterized EuTPT2 and EuTPT4 as E. ulmoides FPS proteins, referred to as EuFPS1 and EuFPS2, respectively[13]. In contrast, the attempt to complement the Δfps lethality with EuTPT1, 3, and 5 was unsuccessful (Supplementary Fig. S2), suggesting that EuTPT1, 3, and 5 did not possess FPS activity. Thus, EuTPT1, 3, and 5 could potentially have a different prenyltransferase activity functionally distinct from FPS.

### EuTPT1, 3, and 5 are novel prenyltransferases.

The function of EuTPT1, 3, and 5 (EuTPTx) was characterized by testing their activity using an in vitro assay. Purified recombinant EuTPTx proteins were incubated with [1-14C]IPP and a series of short-chain allylic substrates, dimethylallyl diphosphate (DMAPP), geranyl diphosphate (GPP, $C_{10}$), farnesyl diphosphate (FPP, $C_{15}$), and geranylgeranyl diphosphate (GGPP, $C_{20}$). The reaction products were analyzed by reverse-phase thin-layer chromatography (RP-TLC; Fig. 1b and Supplementary Fig. S3). All EuTPTx exhibited aberrant polymerization activities; the newly synthesized polyisoprene chains were much longer than the dolichol chains (16–24 IPP units; $C_{80–120}$). The RP-TLC analysis also detected degraded polyisoprene products, which were presumably generated by the oxidative degradation and/or dephosphorylation during the reaction and the following purification of the reaction product in preparation for the RP-TLC analysis (Supplementary Fig. S3b). However, these degradation products indicated that the EuTPTx proteins catalyzed the sequential condensation of IPPs on an allyl substrate, resulting in the formation of much longer-chain polyisoprene than dolichol chains. Most importantly, the predominant reaction products barely developed from the original positions on the RP-TLC plate, suggesting that the EuTPTx proteins are sufficient to produce high molecular weight polyisoprene without any accessory proteins or cofactors. The EuTPTx showed allyl substrate specificities: EuTPT3 and EuTPT5 utilized all allyl substrates as reaction primers, whereas EuTPT1 used GPP, FPP, and GGPP as an allyl substrate, but not DMAPP. However, EuTPT1 had much higher activity than EuTPT3 and EuTPT5 (Fig. 1c and the source data for the main charts is provided in Supplementary Data 1). All EuTPTs showed their activities around neutral pH and mild temperature of 20–40 °C and required divalent cation (Supplemental Table S1).

There are two possible mechanisms for synthesizing TPI. It is possible that the enzymes initially synthesized intermediates of middle- and/or long-chain polyisoprene that they subsequently used as substrates for the next round of sequential IPP condensation. The other possibility is that TPI was produced by rapid non-stop sequential IPP condensation. During the time-course analysis of the EuTPTx reactions, the initial products appeared only at the RP-TLC origin without any short- to long-chain polyisoprene intermediates after 20 min incubation

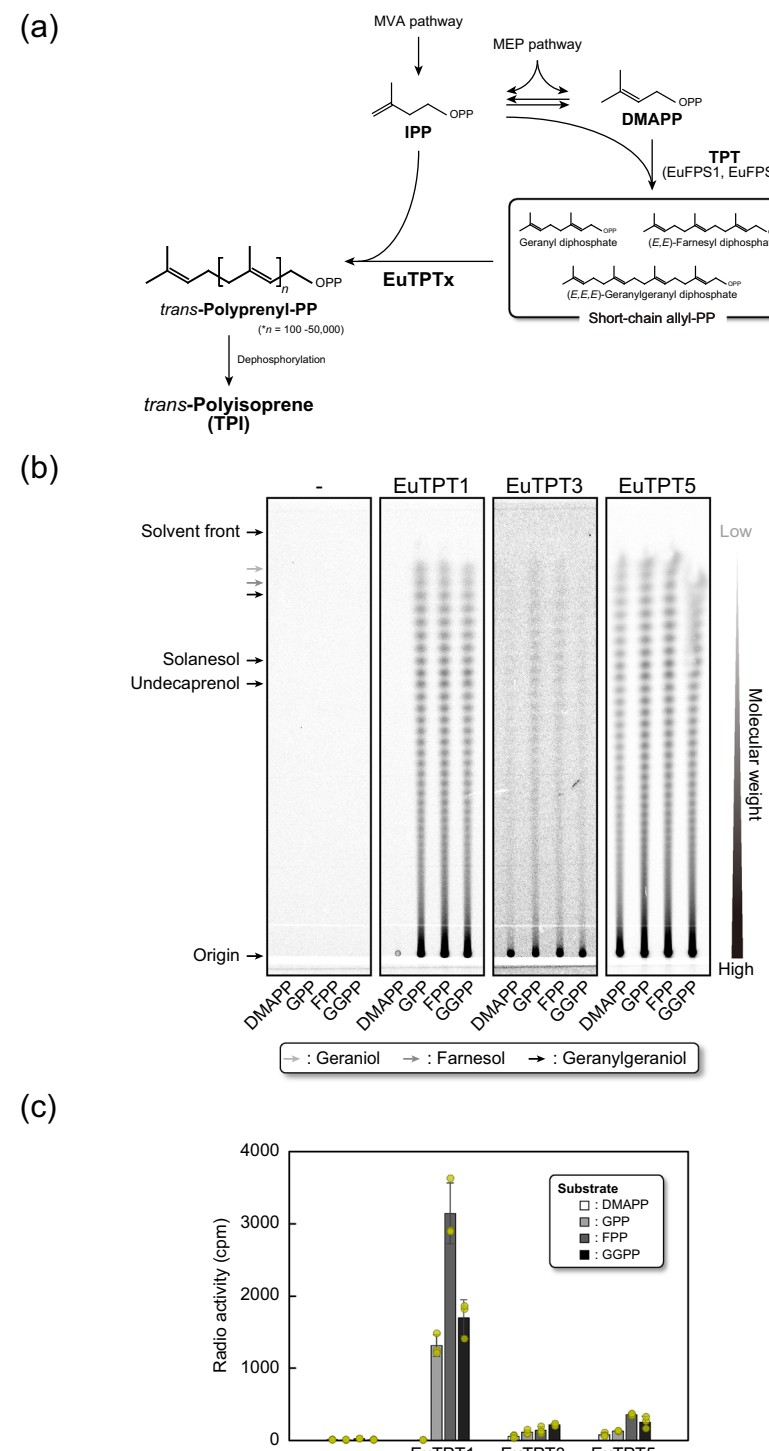

**Fig. 1 Biosynthetic pathway of *trans*-polyisoprene and in vitro analysis of EuTPTx function. a** Flowchart of the biosynthesis pathway of linear *trans*-polyprenyl diphosphate mediated by prenyltransferase. DMAPP dimethyallyl diphosphate, IPP isopentenyl diphosphate, MVA mevalonate, MEP 2-*C*-methyl-D-erythritol 4-phosphate. **b** Reversed-phase thin-layer chromatography (RP-TLC) analysis of reaction products generated by heterologously expressed EuTPTx proteins. [1-$^{14}$C]IPP was incubated with EuTPTx and the allylic substrates, followed by extraction of the $^{14}$C-labeled reaction products, dephosphorylation using acid phosphatase, and separation by RP-TLC. Geraniol ($C_{10}$), farnesol ($C_{15}$), geranylgeranyol ($C_{20}$), solanesol ($C_{45}$), and undecaprenol ($C_{55}$) were used as standards. Geraniol, farnesol, and geranylgeraniol are indicated by light gray, gray, and black arrows, respectively. Origin and solvent front are indicated by black arrows. -: control incubated without any enzyme. **c** Relative activities of EuTPTx using allylic substrates. Allylic substrates were incubated with [1-$^{14}$C]IPP. Unreacted [1-$^{14}$C]IPP was removed and the radioactivities of the $^{14}$C-labeled reaction products were measured using a liquid scintillation counter. All samples were analyzed in independent assays ($n = 3$). Standard deviation was used to calculate error bars. Individual data points are shown in yellow circles.

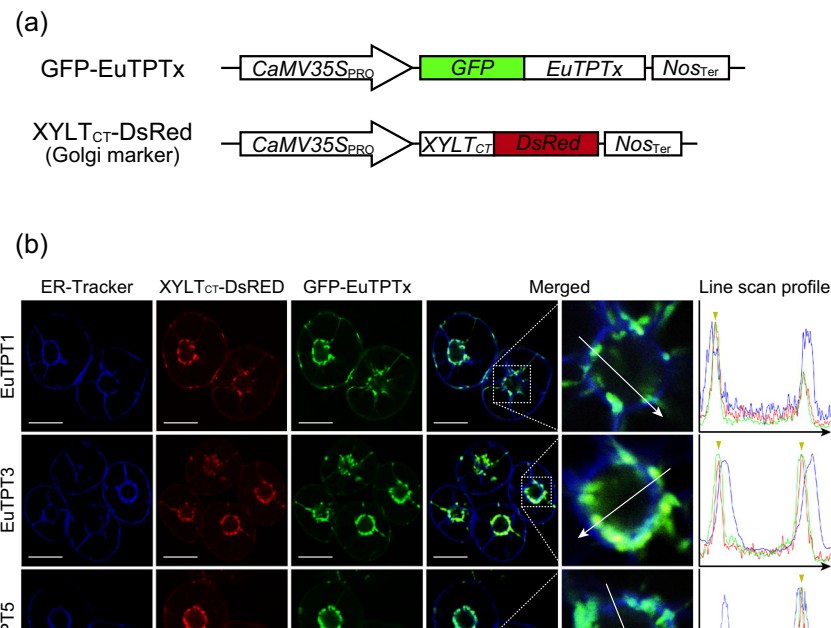

**Fig. 2 Subcellular localization analysis of EuTPTx. a** Schematic representation of chimeric construct GFP-EuTPTx. The coding regions of *EuTPTx* were fused to the C-terminus of *GFP*. XYLT$_{CT}$-DsRed was used as Golgi marker. CaMV35S$_{PRO}$, cauliflower mosaic virus 35S promoter; Nos$_{Ter}$ nopaline synthase terminator. **b** Triple-color imaging by confocal laser scanning microscopy of GFP-EuTPTx-expressing BY-2 cells. Top panels, EuTPT1; middle panels, EuTPT3; bottom panels, EuTPT5. Panels from left to right: ER marker (ER-Tracker), Golgi marker (XYLT$_{CT}$-DsRed), GFP-EuTPTx, and merged images of these three fluorescence signals. (Right) Magnified images of boxed areas around the Golgi/ER/nucleus area are shown. A fluorescence intensity line scan profile was generated along the white arrow in enlarged view, which goes across the Golgi/ER/nucleus area and is shown in the right column. Yellow triangles indicate merged signals of XYLT$_{CT}$-DsRed and GFP-EuTPTx. Blue, ER-Traker; red, XYLT$_{CT}$-DsRed; green, GFP-EuTPTx. White scale bars: 10 μm.

(Supplementary Fig. S4). The signal intensity, *i.e.*, the amount of reaction product, increased in proportion to the reaction time, which was accompanied by the detection of short- to long-chain polyisoprene generated by degradation of the reaction products. These results demonstrated that the EuTPTx enzymes catalyzed the sequential condensation of IPP to produce ultrahigh molecular weight TPI.

EuTPTs have high amino acid sequence similarities with well-characterized plant FPSs, which reside in the cytosol or mitochondria[14,15]. A targeting signal sequence is required for mitochondrial localization, but the EuTPTx proteins did not have any targeting signal sequence, as found in AtFPS1, suggesting that EuTPTx might reside in the cytosol. Another possibility of EuTPTx localization is the endoplasmic reticulum (ER) same as seen in natural rubber-producing CPT[16–18]. In our analysis, EuTPTx was tightly colocalized with the Golgi apparatus (Fig. 2). Line scan profiles of the fluorescent signal intensities from the ER, the Golgi apparatus, and the EuTPTx proteins confirmed that fluorescent signal profiles of EuTPTx absolutely corresponded to those of the Golgi marker, not to those of the ER marker. These results indicated that EuTPT1, 3, and 5 are novel Golgi-localized ultralong-chain prenyltransferases.

**EuTPT3 forms a previously unobserved dimeric architecture.**
To investigate the molecular properties of the EuTPT proteins, we determined the three-dimensional structures of EuTPT3 and EuFPS1 using X-ray crystallography. The crystal structure of apo EuTPT3 was determined from two crystal forms with space groups $P4_32_12$ and $C2$, refined at 3.0 and 3.3 Å resolution, respectively (Table 1). All dimers observed in the two crystal

forms were superimposable with a root-mean-square deviation (RMSD) range of 0.797–0.941 Å over 595–633 aligned Cα atoms. The superimposition parameters suggested that the dimeric architecture of EuTPS3 was highly conserved and unlikely to be caused by crystal packing. Interestingly, the dimeric architecture of EuTPT3 substantially differed from that of EuFPS1 (Fig. 3a, b), which formed a conventional TPT dimer.

Similar to EuFPS1 and other prenyltransferases, the EuTPT3 subunits were composed of 10 core helices (α1-α10) with a deep pocket that included two DDxxD motifs responsible for substrates binding. However, the helices forming the dimer interface were clearly different (Fig. 3c). In EuTPT3, α6 of one subunit interacted with α5 and α6 of the neighboring subunit to form a four-helix bundle (Fig. 3a), whereas, in EuFPS1, α4 of one subunit was inserted between α5 and α6 of the neighboring subunit to form a six-helix bundle of α4–α6 at the dimer interface (Fig. 3b). As a result, the neighboring subunit of EuFPS1 was tilted at an angle of ~45° compared with that of EuTPT3 (Fig. 3c). Thus, EuTPT3 had a larger central tunnel than EuFPS1 (Fig. 4a, b). The tunnel is mainly composed by hydrophobic residues of α5 and α6 (Fig. 4c), which is considered the path for the products because the chain length determination element locates in this region.

Among the structural elements involved in substrate binding and catalytic activity, EuTPT3 featured two conserved DDxxD motifs at the active site (Fig. 5a), both of which could contribute to substrate recognition. The crystal structures of *Escherichia coli* FPS complexed with IPP and *Trypanosoma crusi* FPS complexed with DMAPP show that the substrate phosphate groups are recognized by basic residues in the active site[19,20]. EuTPT3 also featured structurally conserved basic residues (Lys46, Arg50, Arg109, and Lys348) in the equivalent positions (Supplementary

**Table 1 Data collection and refinement statistics.**

| | *Eu*TPT3 WT form 1 | *Eu*TPT3 WT form 2 | *Eu*TPT3 (C94Y/A95F) | *Eu*FPS1 |
|---|---|---|---|---|
| Data collection | | | | |
| Space group | $P4_32_12$ | $C2$ | $P3_221$ | $P2_1$ |
| Cell dimensions | | | | |
| $a, b, c$ (Å) | 80.26, 80.26, 278.95 | 234.39, 131.06, 103.92 | 84.25, 84.25, 181.19 | 54.00, 71.27, 103.38 |
| $\alpha, \beta, \gamma$ (°) | 90, 90, 90 | 90, 111.98, 90 | 90, 90, 120 | 90, 103.50, 90 |
| Resolution (Å) | 44.0–3.0 (3.11–3.00)[a] | 39.0–3.3 (3.42–3.30) | 42.1–3.3 (3.42–3.30) | 33.6–2.2 (2.28–2.20) |
| $R_{merge}$ | 16.1 (>100) | 8.3 (97.4) | 33.1 (>100) | 17.0 (>100) |
| $I / \sigma I$ | 24.70 (2.20) | 14.96 (1.97) | 12.20 (2.48) | 10.77 (3.01) |
| Completeness (%) | 99.87 (99.84) | 99.82 (99.95) | 99.90 (99.91) | 98.68 (98.07) |
| Redundancy | 28.6 (29.5) | 5.8 (5.8) | 21.1 (22.0) | 7.5 (7.7) |
| Refinement | | | | |
| Resolution (Å) | 44.0–3.0 (3.10–3.00) | 39.0–3.3 (3.42–3.30) | 42.1–3.3 (3.42–3.30) | 33.6–2.2 (2.28–2.20) |
| No. reflections | 547,833 (54,592) | 253,935 (25,685) | 248,162 (25,008) | 287,166 (29,298) |
| $R_{work}/R_{free}$ | 0.223/0.233 | 0.277/0.301 | 0.254/0.312 | 0.226/0.244 |
| No. atoms | | | | |
| Protein | 5482 | 18,385 | 5106 | 5441 |
| Ligand/ion | 0 | 0 | 0 | 0 |
| Water | 0 | 0 | 0 | 63 |
| *B*-factors | – | – | – | – |
| Protein | 88.31 | 113.96 | 65.42 | 57.93 |
| Ligand/ion | – | – | – | – |
| Water | – | – | – | 40.80 |
| R.m.s. deviations | | | | |
| Bond lengths (Å) | 0.011 | 0.004 | 0.004 | 0.005 |
| Bond angles (°) | 1.24 | 0.78 | 0.78 | 0.80 |

[a]Values in parentheses are for highest-resolution shell.

Fig. S5). These observations indicated that TPT and FPS share the same catalytic mechanism. However, the α4-α5 loop of EuTPT3 and the substrate-bound FPS adopted different conformations (Fig. 5a, b), suggesting that substrate binding proceeds via an induced fit.

**Structural comparisons of EuTPT3 and EuFPS1.** The subunit structure of EuTPT3 was mostly similar to that of EuFPS1 (RMSD Cα = 1.672 Å over 300 aligned Cα atoms). Superimposition showed that the α1–α3 and α6–α10 regions were aligned, but distinctive deviations were observed in the α4–α5 region (Fig. 5a, b). Because α4–α6 were located at the dimer interface, the deviations affected the dimeric architecture.

The structural difference is related to the chain length determination residues, which are conserved as bulky amino acids among FPSs, including EuFPS1 and 2. Specifically, the equivalent positions of residues Cys94 and Ala95 on α4 in EuTPT3 are occupied by Tyr88 and Phe89 in EuFPS1. The side chains of those residues protrude between α5 and α6, resulting in a marked bending of α5 (Fig. 5c). The residues equivalent to Tyr88 and Phe89 in EuFPS1 are highly conserved among all FPSs. The residues equivalent to Cys94 and Ala95 in EuTPT3 are conserved in the EuTPT1 and EuTPT5 sequences. Therefore, those residues were predicted to be critical for the α5 arrangement and dimeric architecture. To test this hypothesis, we replaced Cys94 and Ala95 in EuTPT3 with tyrosine and phenylalanine, respectively, to create the double mutant EuTPT3(C94Y/A95F). An in vitro test demonstrated that the double mutant lacked the activity for synthesizing ultrahigh molecular weight TPI (Supplementary Fig. S6). Interestingly, we determined the crystal structure of the EuTPT3(C94Y/A95F) mutant protein at 3.3 Å resolution (Table 1) and found that it formed the FPS dimeric conformation (Supplementary Fig. S7). The mutated residues changed the orientation of α5, compared with that in the wild type. The RMSD values of the double mutant subunit compared

to the wild-type structures of EuTPT3 and EuFPS1 were 3.41 Å (no. of Cα = 202) and 1.85 Å (no. of Cα = 284), respectively. Thus, the replacement of only two residues switched the dimeric conformation from the TPT-type to the FPS-type dimer. Actually, the replacements of the exact two amino acids in other EuTPTs and their endo product analysis provided reasonable results; mutations of Cys94 and Ala95 in EuFPS1 and Cys96 and Ala96 in EuTPT5 to Tyr and Phe, respectively, resulted in synthesizing FPP, whereas Tyr88 and Phe89 in EuTPT2 (EuFPS1) and Phe95 and Phe96 in EuTPT4 (EuFPS2) produced longer chain length end-product than FPP (Supplementary Fig. 8). These results indicated that Cys94 and Ala95 on α4 in EuTPT3 and the corresponding amino acid in other EuTPTs were essential for the TPT-dimer conformation, which forms the path for newly synthesized ultrahigh molecular weight TPI.

**Spatiotemporal analysis of ultrahigh molecular weight TPI biosynthesis in plant.** Transgenic tobacco plants expressing *EuTPTx* were generated to determine the in vivo activity of EuTPTx and its contribution to the biosynthesis of TPI (Supplementary Fig. S9). The accumulation of TPI *in planta* with a high expression level of *EuTPTx* was observed by analyzing stem sections stained with Nile red (Supplementary Fig. S9c). The Nile red dye uniquely stains TPI due to the solvatochromic effect[21,22]. Transgenic plants, which expressed *EuTPTx*, hardly accumulated newly synthesized TPI but exhibited specific Nile red-positive signals around vessels only, whereas wild-type plants were almost negative to Nile red staining. Moreover, the wavelength with the highest signal intensity in the fluorescence spectrum corresponded to that of TPI, indicating that EuTPTx synthesized polyisoprene *in planta*.

The fruit, especially the pericarp, is the best TPI-producing part in *E. ulmoides* plants[9], and, thus, suitable for analyzing TPI biosynthesis *in planta*. To confirm the correlation between TPI production and the presence of EuTPTx, we subjected *E. ulmoides*

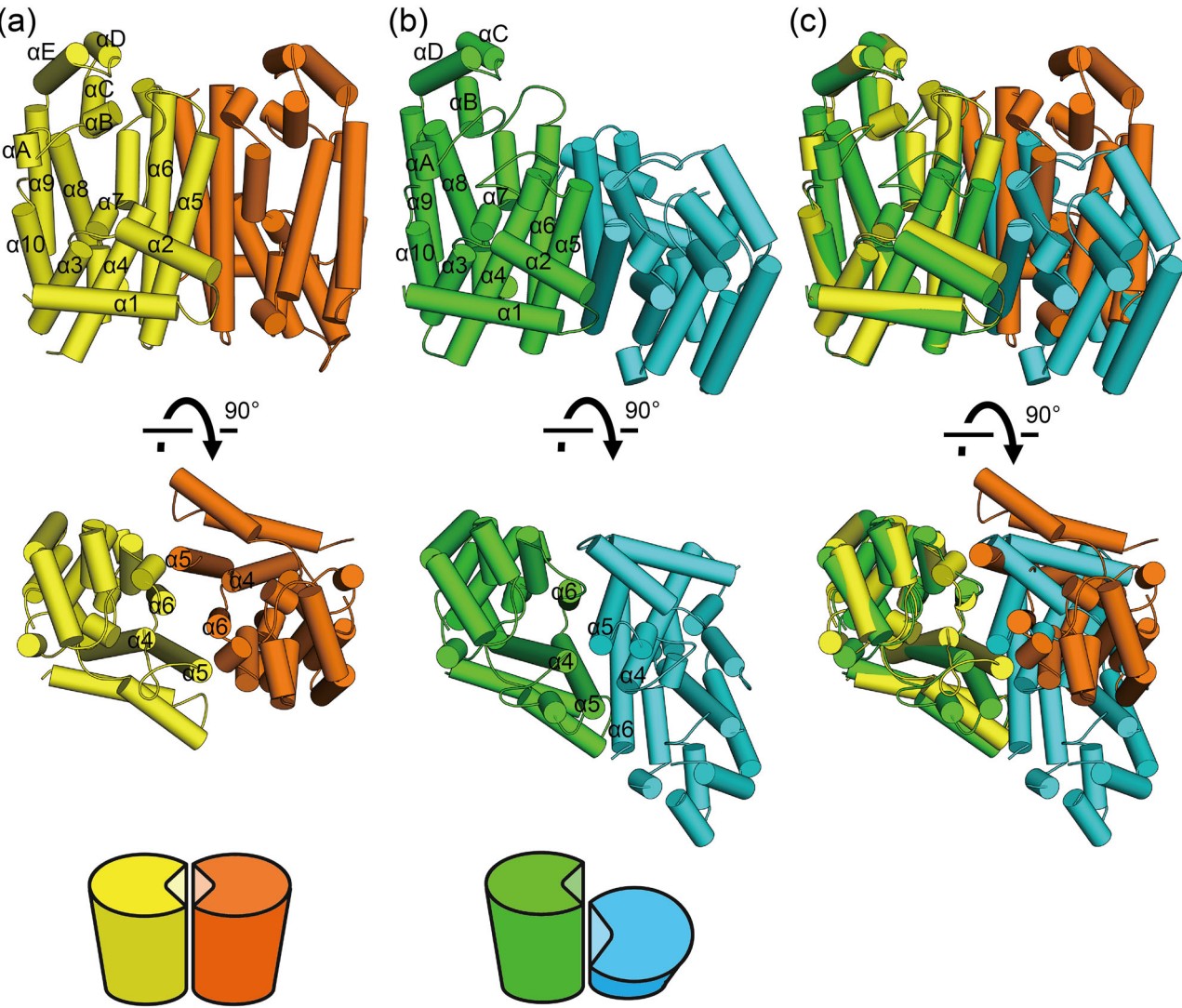

**Fig. 3 Dimeric structures of EuTPT3 and EuFPS1.** Graphic representations of overall structures of EuTPT3 (**a**) and EuFPS1 (**b**) are shown. The helices are represented as cylinders. Chain A and B of EuTPT3 are colored in yellow and orange, respectively. Chain A and B of EuFPS1 are colored in green and cyan, respectively. Simplified models of the dimeric architectures are shown in bottom. **c** Superimposed structures of EuTPT3 and EuFPS1 based on chain A.

fruits to a spatiotemporal analysis focused on morphological features, *EuTPTx* expression levels, and TPI accumulation. Fruit development in *E. ulmoides* includes two steps. Firstly, the fruit increases in size, and, secondly, the fruit's maturation is accompanied by an increase in its weight (Supplementary Fig. S10). Remarkably, the *EuTPT5* transcript level increased gradually during the first fruit development step, resulting in a substantial increase, which corresponded completely to the accumulation of TPI in the pericarp (Fig. 6a and the source data is provided in Supplementary Data 1). Histochemical analysis of TPI in the fruit also demonstrated a strong correlation between *E. ulmoides* fruit development, *EuTPT5* expression, and TPI accumulation. The slow accumulation of TPI was observed during the first fruit developmental step, but a substantial increase in TPI biosynthesis and accumulation was detected as the fruits progressed from the first to the second developmental step (Supplementary Figs. S10 and S11). The TPI content in the pericarp hardly changed after reaching the highest plateau despite a decreasing *EuTPT5* expression level during the fruit maturing step. This restricted expression peak of *EuTPT5* was repeated every year, whereas the expression level of *EuTPT1* and *EuTPT3* varied much less than that of *EuTPT5* (Supplementary Fig. S12).

Furthermore, during TPI production, only the EuTPT5 protein was detected in TPI-accumulating cells along the luminal side of the fruit (Fig. 6b). These results indicated that the presence of EuTPT5 was closely related to the TPI biosynthesis in *E. ulmoides* pericarp.

In natural rubber biosynthesis, the CPT and the required cofactor proteins are localized at the surface of rubber particles[23–25]. This suggested that prenyltransferases involved in ultrahigh molecular weight TPI biosynthesis might be localized at the surface of polyisoprene aggregates. Centrifugation of a pericarp homogenate separated a creamy floating fraction containing ultrahigh molecular weight polyisoprene (Mw: $5.0 \times 10^6$; Supplementary Fig. S13), which was similar in appearance as fractions isolated from rubber-producing trees. Further analysis of proteins tightly bound to TPI detected EuTPT5 but not EuTPT1 or EuTPT3 (Supplementary Fig. S14). Moreover, in TPI-producing pericarp cells, the localization of EuTPT5 was specific for polyisoprene aggregates, whereas the protein was hardly detected as cytosolic or microsomal protein. Similar to the result of the *EuTPT5* expression analysis, larger amounts of EuTPT5 protein were detected together with polyisoprene during the TPI production period (Fig. 6c). Furthermore, the specific association between EuTPT5 and polyisoprene

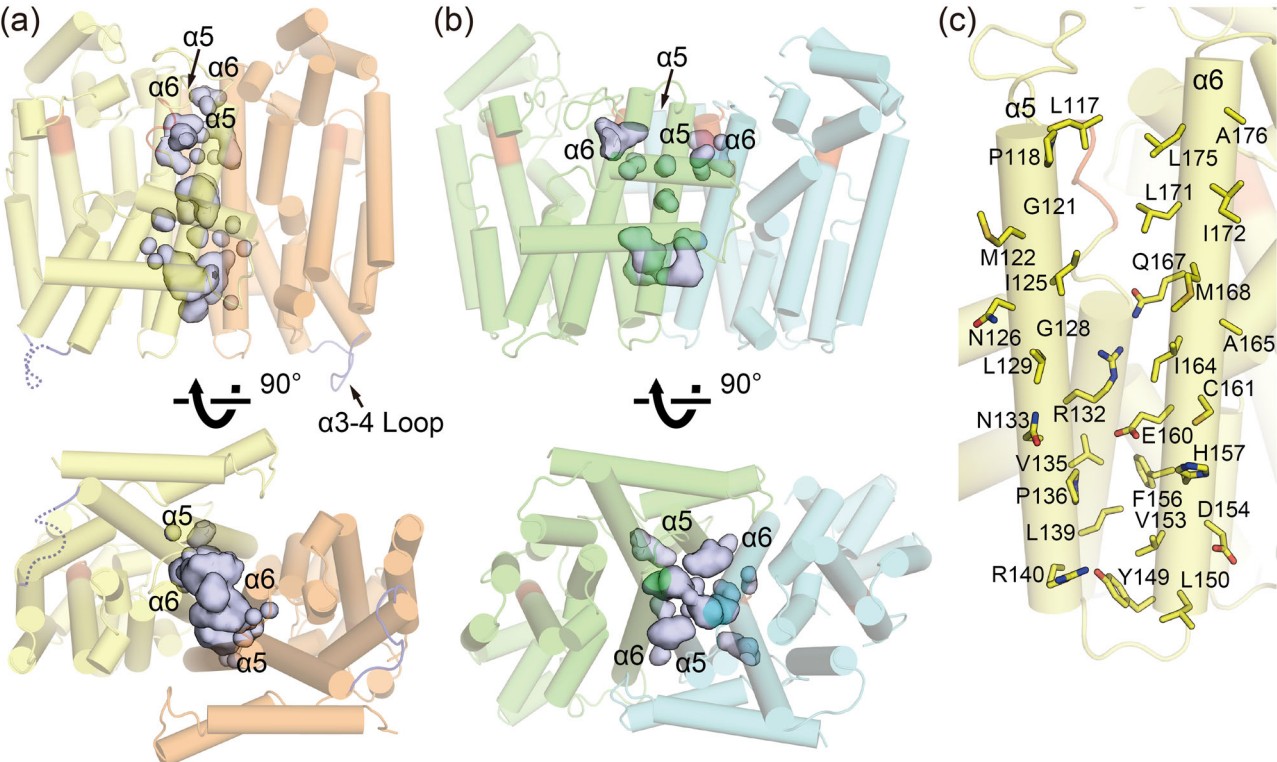

**Fig. 4 Central cavities of EuTPT3 and EuFPS1.** Side and bottom views of **a** EuTPT3 and **b** EuFPS1 with the central cavities shown as gray surface model. Cavities were generated by PyMOL. Cartoon ribbon models of EuTPT3 and EuFPS1 are shown transparently. The DDxxD motif regions are colored in red. The α3–4 loops of EuTPT3 are colored in dark blue. The dashed line represents the untraceable α3–4 loop in chain A of EuTPT3. **c** Amino acids constructing the central cavity of EuTPT3.

was similar to that between the required rubber biosynthesis proteins and the rubber particles[17,24,25], i.e., EuTPT5 was detected across the surface and/or around TPI aggregates, whereas it barely colocalized with the polymeric product (Fig. 6d). These results indicated that EuTPT5 was the main contributor to in vivo TPI biosynthesis.

## Discussion

The rapid expansion of genomic information and the progress in instrumental analyses have facilitated the identification of uncharacterized prenyltransferase activities and their key determinants leading to end product preferences[26–28]. However, these studies also suggest that the physiological function of many prenyltransferases is still uncertain. Indeed, the occurrence of TPI in a few plants suggests the presence of unidentified TPT(s) that differ substantially from FPS. A previous study, which examined genes potentially involved in TPI biosynthesis, identified EuTPT5 as a candidate for this function[12]. However, the study's RNA-Seq results were controversial because the highest expressed candidate gene during TPI accumulation was EuTPT1[12]. The study also indicated that all EuTPTs were localized at the ER, which included FPS-type EuTPT2 and 4[13] that are likely to reside in the cytosol or mitochondria because these are the only cellular subcompartments known to contain FPSs in plants[14,15]. Thus, TPI biosynthesis and accumulation remained unclear. In this study, we identified three novel functional ultralong-chain TPTs, EuTPT1, 3, and 5, using a recently established *E. ulmoides* EST library and characterized their activities with biological and histological approaches, complemented by enzymological and structural analyses. These prenyltransferases were Golgi-localized enzymes that catalyzed the non-stop condensation of large

numbers of IPP molecules to the allyl substrate without any accessory proteins. We further demonstrated that especially EuTPT5 was indispensable for TPI biosynthesis and accumulation in *E. ulmoides*. Importantly, the three-dimensional structure of EuTPT3 differed substantially from that of the FPSs despite the high amino acid similarities shared between EuTPTx and FPSs.

Our structural analysis identified a unique, TPT-type dimer conformation, in which the central tunnel connected the active site with the bottom of the molecule. This dimeric architecture is described for other *trans*-prenyltransferases deposited in the Protein Data Bank (PDB; http://www.rcsb.org/pdb/)[29]. Interestingly, the structural comparison of EuTPT3 and EuFPS1 identified a few TPT-unique amino acids that stabilized the distinct conformation of the subunits with a different overall orientation of one subunit within the dimer, compared with that in FPS-type proteins. These residues, conserved among EuTPTx, stabilized close-packing interactions between α5 and α6 (Fig. 5a–c). We even succeeded in switching the EuTPT3 dimeric conformation to the FPS-type dimer by mutating only two residues, Cys94 and Ala95, to Tyr and Phe, respectively. Thus, those two residues not only play a key role in determining the product length but also in the formation of the TPT-type dimer.

A previously reported structural analysis of *E. coli* octaprenyltransferase showed that bulky amino residues on α5 and α6 (Met123 and Met135) were determinants for preventing the formation of a longer products[30]. The equivalent bulky amino residues were conserved in the α4–6 region of not only EuTPTx but also EuFPS1 and 2. Thus, the mechanism facilitating TPI synthesis is likely to require a fundamentally different molecular architecture of the TPT dimer. The TPT-type dimer formed a larger hollow at the dimer interface and supported a hydrophobic tunnel reaching through the bottom region (Fig. 4a, b), whereas the bulky side

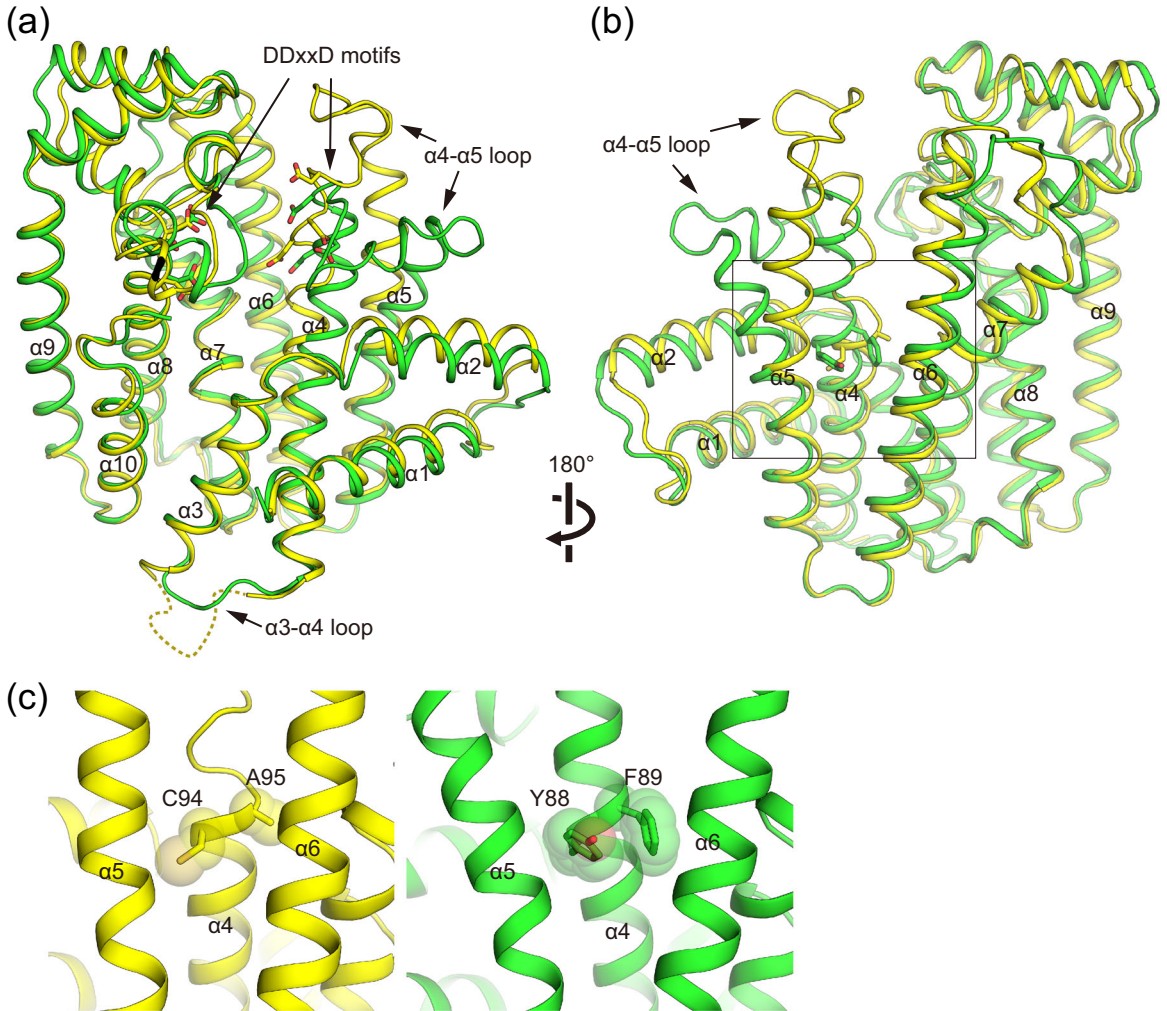

**Fig. 5 Structural comparisons of EuTPT3 and EuFPS1. a** Ribbon drawings of superimposed subunit structures of EuTPT3 (yellow) and EuFPS1 (green) are shown. DDxxD motifs are represented as stick model. The untraceable α3–4 loop in chain A of EuTPT3 is represented as dashed line. **b** Side view of dimer interface. Cys94-Ala95 of EuTPT3 and Tyr88-Phe89 of EuFPS1 in α4 helix are represented as stick model. **c** Close-up views around residues Cys94-Ala95 of EuTPT3 (left) and Tyr88-Phe89 of EuFPS1 (right).

chains of Tyr88 and Phe89 appeared to prevent the formation of a tunnel in EuFPS1. The double mutant F112A/F113S of avian FPS synthesized polyisoprene with increased chain length (<$C_{50}$)[7]. Furthermore, EuTPT3 conserved a hydrophobic hollow in the bottom region of the dimer interface (Fig. 4). These observations suggested that the longer polyisoprene products may pass through the hydrophobic tunnel at the dimer interface. Although the relationships among sequences, structures, and products of FPS have been well studied[29], producing TPI (e.g., longer than $C_{100}$) using an FPS mutant has never been achieved. Here, we found that the dimeric architecture of EuTPT3 featured a wider tunnel at the dimer interface. Thus, our structural analysis could pave the way for studying the mechanism of TPI biosynthesis. Among the TPI synthases, EuTPT3 shares 77 and 72% amino acid identity with EuTPT1 and EuTPT5 (Supplementary Fig. S15). Although the most residues around the DDxxD motifs are structurally conserved, EuTPT3 has lower activity than EuTPT1 (Fig. 1c). Residues Ile164-Ala165 on α6 in EuTPT3 are occupied by Val164-Gly165 in EuTPT1 and Val165-Ala166 in EuTPT5, respectively (Supplementary Fig. S15). Crystal structure of EuTPT3 showed that the residues Ile164-Ala165 contact with Ile164 and Ile125 of the neighboring subunit in the hydrophobic tunnel, suggesting that smaller side chains such as valine and glycine provide the wider path for the

product that may contribute to the activity difference (Fig. 4c and Supplementary Fig. S16).

More than 2500 higher plant species are known to produce natural rubber[31–33], which is composed mostly of ultrahigh molecular weight polyisoprene with *cis*-configuration, indicating that it is synthesized by CPT[16,23,34]. The CPTs in rubber biosynthesis depend on accessory proteins, such as rubber elongation factor, small rubber particle protein (SRPP), and a family of Nogo-B receptor, for catalysis[16,17,24,25,35,36], i.e., there is no CPT with an intrinsic activity that can work without forming a heteromeric complex. Natural rubber is typically synthesized inside of the ER bilayer membrane in the presence of phospholipid, especially phosphatidylcholine. Natural rubber particles enclosed by a lipid monolayer are budded off from the ER with the accessory proteins on the surface[16,24,25,35,36]. This indicates that CPT is initially localized at the ER. The surface of natural rubber particles is covered with negatively charged proteins, including SRPP, and lipids, which contribute to the negative zeta potential and colloidal stability that prevents the formation of fusions among rubber particles[25,37]. In contrast, our study demonstrated that TPTs alone are sufficient to elongate the chain length of an isoprene unit. Furthermore, EuTPTx resided at the Golgi apparatus, and TPI further formed

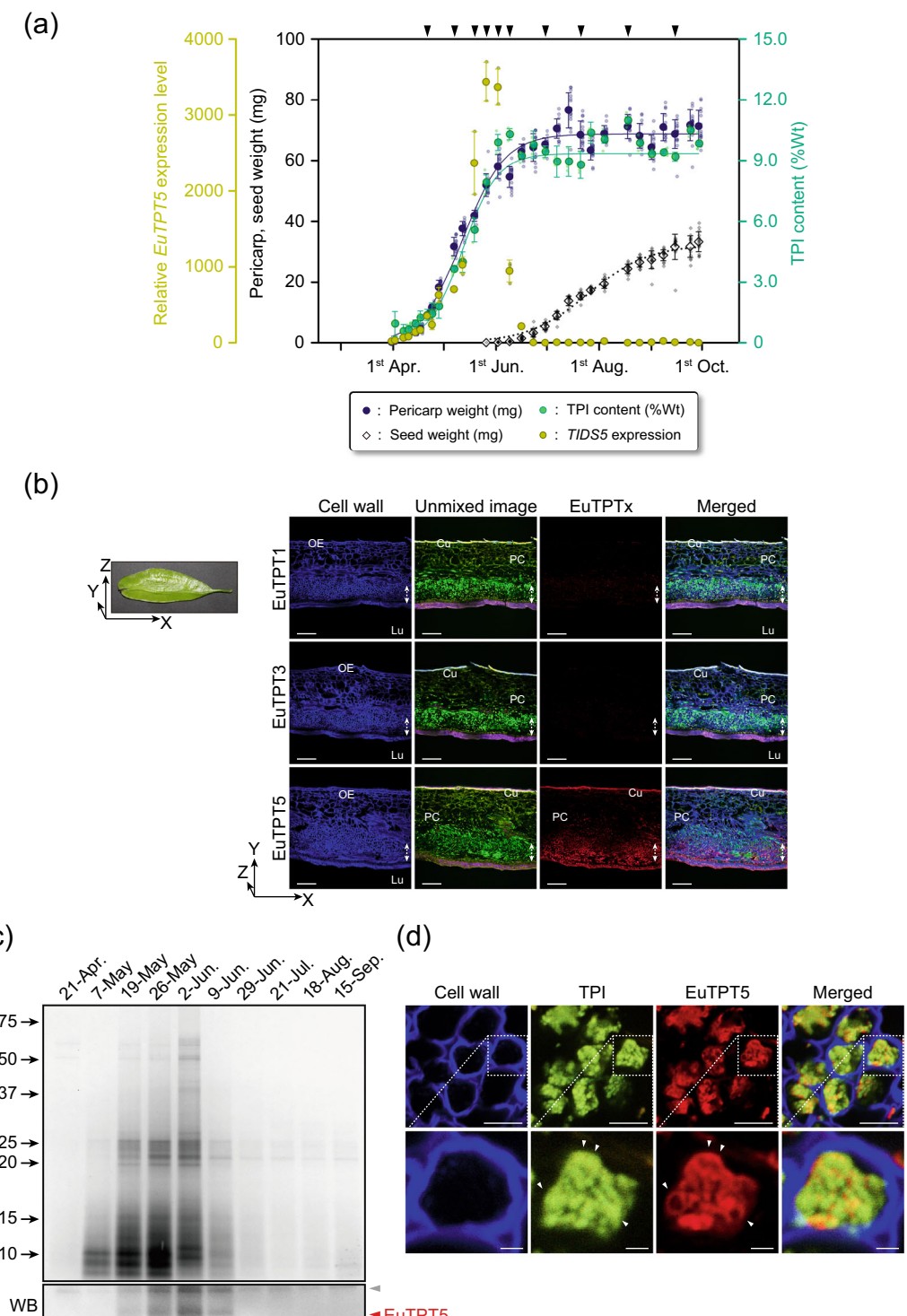

a fibrous structure due to the fusion of TPI granules[22]. Some plant prenyltransferases form heterodimeric protein complexes[16,38–40], consisting of a catalytic subunit associated with a non-catalytic subunit that typically lacks some essential domain(s) for the catalytic activity but promotes catalytic fidelity and activity of the complex. However, it is still uncertain whether TPI biosynthesis by EuTPTx involves complex formation with other subunits, such as ER-localized, CPTs' putative SRPP homologs, EuSRPP1, 2, and 7[12]. Furthermore, bacterially produced EuTPTx alone was sufficient for TPI synthesis, which is different from CPTs that require cofactor proteins responsible

for the biosynthesis of dolichols and/or long-chain polyisoprene ($>C_{70}$)[41–43]. Moreover, the analysis using non-rubber producing plants expressing EuTPTs in this study revealed that EuTPTs do not require any additional proteins for their activities and localizations. Therefore, EuSRPP1, 2, and 7 are not considered as accessory proteins involved in TPI biosynthesis. Here, we identified the structural key elements responsible for maintaining a dimeric architecture of EuTPTx that differs substantially from that of other prenyltransferases. However, unraveling the process of subcellular targeting of EuTPTx will further elucidate the underlying mechanism for the biosynthesis

**Fig. 6 TPI accumulation in *E. ulmoides* pericarp. a** Correlation of *EuTPT5* expression (closed yellow circle) with pericarp weight (closed dark blue circle), seed weight (open diamond), and accumulation of TPI (closed green circle) in 2015. Logarithmic growth curves for pericarp weight, seed weight, and accumulation of TPI are represented as solid dark blue line, dotted line, and solid green line, respectively. Regression equations are as follows: pericarp weight (mg), $Y = 68.76/[1 + 33.07 \ast \exp^{-0.08 \ast (\text{Date from 31 March})}]$, $r^2 = 0.9747$; Seed weight (mg), $Y = 32.30/[1 + 715.12 \ast \exp^{-0.06 \ast (\text{Date from 31 March})}]$, $r^2 = 0.9612$; TPI content (%Wt), $Y = 9.35/[1 + 73.29 \ast \exp^{-0.10 \ast (\text{Date from 31 March})}]$, $r^2 = 0.9651$. All $r^2$ values of pericarp weight and seed weight and TPI content are above 0.9. Pericarp weight and TPI content reached equilibrium around June, whereas the seed weight reached equilibrium around September. Black triangles at the top indicate the sampling date for further protein analysis. All analysis of *EuTPT5* expression and TPI content were performed in independent samples ($n = 3$). The averages of pericarp weight and seed weight ($n = 13$–15) are shown in closed dark blue circle and open diamond, respectively. **b** Histochemical and immunostaining of *E. ulmoides* pericarp. (Left) The presentation of fruit orientation used for preparing the sections. (Right) SCLSM analysis of pericarp sections of XY direction. From left to right panels: cell wall staining, unmixed imaging of Nile red staining, immunostaining using anti-EuTPTx antibody, and merged images of these fluorescence. White arrows indicate TPI accumulation region. OE outer epidermis, Lu lumen, PC parenchyma cells, Cu cuticle. White scale bars: 100 μm. Direction of the section is shown in left bottom. **c** Expression profile of EuTPT5. (Top) Silver staining of total TPI proteins. (Bottom) Western blotting analysis of TPI proteins using an anti-EuTPT5 antibody. Red and gray triangles indicate EuTPT5 and non-specific band, respectively. The proteins were extracted from 0.5 mg TPI on 21-April, 2 mg TPI on 7-May, and 5 mg TPI on the remaining dates. **d** Localization analysis of EuTPT5. (Top, left to right panels) Cell wall staining, unmixing images of TPI, immunostaining of EuTPT5, and merged images. White scale bars: 10 μm. (Bottom) Enlarged images of cell wall staining, unmixing images of TPI, and immunostaining of young epithelial cells. Bottom images are close-up views of the upper images. White triangles indicate TPI or EuTPT5. White scale bars: 2 μm.

and accumulation of TPI *in planta*, which is a topic of great interest for future investigations.

## Methods

**Materials**. Nile red and Fluorescent Brightener 28 were purchased from Sigma (St. Louis, MO), and ER-tracker™ Blue-White DPX was purchased from Molecular Probes (Eugene, OR). Anti-EuTPTs antibodies were obtained from Sigma or GeneDesign (Osaka, Japan) using polyclonal antibody production services. The polyisoprene standard, Polyisoprene (1,4-addition) (Mw: 1,314,500; Mn: 1,194,400), was purchased from Polymer Source, Inc. (Dorval, Quebec, Canada).

**Plant materials**. *E. ulmoides* fruits were collected from field-grown trees in Osaka, Japan, every week from 2013 to 2015. The samples were stored at $-80\,^{\circ}\text{C}$ or placed in a mountant before storage at $-80\,^{\circ}\text{C}$ until analysis.

**Cloning of EuTPTs and construction of expression vectors**. The preparation of cDNA from *E. ulmoides* leaves was initiated by isolating mRNA using the RNeasy Plant Mini Kit (QIAGEN, Chatsworth, CA) and treating it with DNase (NEB, Beverly, MA) to eliminate genomic DNA contamination. RNA aliquots (1.0 μg) were reverse-transcribed using a Superscript VILO kit (Invitrogen, Carlsbad, CA). DNA fragments containing putative *EuTPTs* were amplified using KOD plus Neo polymerase (TOYOBO, Osaka, Japan) and specific primer sets (Supplementary Data 2), subcloned, and sequenced. Yeast, plant, and *E. coli* expression vectors were constructed using restriction enzyme-digested *EuTPTs* as inserts for the pYES2 vector (Invitrogen, Carlsbad, CA), pBI121 vector, and pCold I vector (TaKaRa, Shiga, Japan), respectively.

**Complementation of the *S. cerevisiae* Δfps mutant**. The following *S. cerevisiae* strain was provided by the National Bio-Resource Project (NBRP) of the MEXT, Japan: BY4743 (*MATa/α his3Δ1/his3Δ1 leu2Δ0/leu2Δ0 lys2Δ0/LYS2 met15Δ0/MET15 ura3Δ0 /ura3Δ0*). The *S. cerevisiae* erg20 heterozygous knockout strain (*MATa/α his3Δ1/his3Δ1 leu2Δ0/leu2Δ0 lys2Δ0/LYS2 met15Δ0/MET15 ura3Δ0/ura3Δ0 Δfps::KanMX/FPS*) was purchased from Thermo Fisher Scientific (Waltham, MA).

The yeast expression plasmids pYES2-*EuTPTs* or pYES2-*ScFPS* were introduced into the yeast Δfps heterozygous diploid mutant by electroporation. The resultant transformants were cultivated on sporulation medium (1% potassium acetate, 0.1% yeast extract, and 0.5% glucose), and tetrad analysis was performed.

**Heterologous EuTPTs expression and purification**. The *E. coli* expression vectors pCold and pCold-*EuTPTs* were introduced into *E. coli* JM109. The transformant cells were cultivated in 2 mL of 2 × YT medium (1% yeast extract, 1.6% Bacto peptone, 0.5% NaCl) containing 50 μg/L carbenicillin overnight. The pre-culture was transferred into 100 mL of the same medium and further incubated to an $OD_{600}$ of 0.5. The culture was placed on ice for 30 min. Then, IPTG induction was initiated by adding 1 mM IPTG, and the cells were further cultivated for 20 h at 15 °C. The cells were collected by centrifugation at 4 °C, $2500 \times g$ for 5 min, and then suspended in 50 mM Tris-HCl buffer pH 7.5, 500 mM NaCl, and 5 mM imidazole (buffer A). The resuspended cells were disrupted by sonication on ice, followed by centrifugation at 4 °C, $20,000 \times g$ for 5 min to remove cell debris. The supernatant was used as the total soluble protein.

The recombinant proteins were purified from the total soluble protein using a TALON® Metal affinity column (TaKaRa). After washing with buffer A, each recombinant enzyme was eluted with buffer A containing 100 mM imidazole. The

eluted enzymes were concentrated, and the buffer was exchanged using an Amicon® Ultra-0.5 mL 30 K. The concentrations of the purified proteins were measured by Bradford assay. The total soluble protein and the purified proteins were separated by SDS-PAGE using 5–20% gradient gels (Wako, Osaka, Japan), and the protein bands were visualized by CBB staining or western blotting using anti-EuTPTs antibodies.

**EuTPTs assay**. Assays for purified EuTPTs using radiolabeled [1-$^{14}$C]IPP was carried out in a 100-μL total reaction volume containing 50 mM cacodylic acid, pH 7.0, or 50 mM Tris-HCl, pH 7.5, 5 mM $MgCl_2$, 0.1% Triton X-100, 20 μM [1-$^{14}$C] IPP, supplemented with 100 μM DMAPP, GPP, FPP, or GGPP, and 1.0 μg of purified protein. Reaction samples were incubated at 25 °C for 20 h and terminated by adding 1 mL chloroform/methanol mixture (2:1, v/v), followed by vortexing. The radiolabeled products were extracted with a chloroform/methanol/water mixture (3:48:47, v/v/v), and the organic layer was recovered. The radioactivity was measured using a Beckman LS6500 multi-purpose scintillation counter (Beckman Dickinson and Co., Sparks, MD). The radiolabeled reaction products were dephosphorylated using acid phosphatase from potato (Sigma) as previously reported[13] with some modifications: the chloroform/methanol/water extracted products were dried and mixed with a final concentration of 10 mM acetate buffer, pH 4.8, 0.1% Triton X-100, 60% methanol, and 1.5 units of acid phosphatase in 100 μL for incubation at 37 °C for 20 h. The hydrolyzed products were extracted with *n*-hexane, spotted onto reversed-phase thin-layer chromatography (RP-TLC) plates, Silica gel 60 RP-18 $F_{254}$S (Merck Millipore, Tokyo, Japan), and separated using an acetone/water/hexane mixture (199:1:20) or an acetone/water mixture (9:1, v/v) as solvent. The position of the standard prenyl alcohols was visualized by exposing the dried plate to iodine vapor. The RP-TLC-separated products were visualized by autoradiography using a Typhoon FLA7000 laser scanner (GE Healthcare, Tokyo, Japan).

**Generation of transgenic plants**. Leaf disks of wild-type *Nicotiana tabacum* SR-I plants were co-cultivated with *Agrobacterium tumefaciens* LBA4404 strain carrying pBI121-*EuTPTs* in the dark for 3 days on callus inducing medium (CIM): Murashige and Skoog (MS) medium supplemented with 30 g/L sucrose and 0.5 g/L MES, pH 5.2, 0.2 mg/L benzylaminopurine, 2 mg/L 1-naphthaleneacetic acid, and 3 g gellan gum. The disks were placed onto CIM containing 50 mg/L kanamycin, 250 mg/L carbenicillin for cultivation at 25 °C with a 16-h photoperiod; the plates were changed every week. Then, an individual shoot was separated and transferred onto hormone-free medium containing 50 mg/L kanamycin, 250 mg/L carbenicillin for selection. The regenerated plantlets (designated the $T_1$ generation) were transferred to soil and maintained under greenhouse conditions. The $T_2$ and $T_3$ generation of transformants were used for real-time PCR analysis and histochemical analysis, respectively.

**Subcellular localization analysis**. All fusion constructs, *GFP-EuTPTx*, were generated by PCR methods using full-length *EuTPTs* and *GFP* as templates and ligated with the pBI121 vector. The resultant vectors were introduced into *Agrobacterium tumefaciens* LBA4404 by electroporation. Tobacco BY-2 cells stably expressing Golgi marker $XYLT_{CT}$-DsRed[44,45] were generated using pGPTV-HPT-$XYLT_{CT}$-DsRed and by Agrobacterium-mediated transformation[45] and super-transformed with the GFP fusion constructs. Fluorescence signals were documented 3–4 days after the sub-cultivation. Cells expressing GFP and DsRed fusion proteins were first stained with 1 μM ER-tracker and analyzed with DIGITAL ECLIPSE C1si (Nikon, Tokyo, Japan) equipped with CFI Plan Apo objectives and EZ-C1 3.40 software (Nikon). Fluorescence was excited with the 408-nm line of a

blue diode laser, the 488-nm line of a solid laser, and the 543-nm line of a G-HeNe laser.

### Histochemical analysis and immunostaining of fruit tissue samples.
Approximately 20 μm-thick cryosections of *E. ulmoides* fruit were prepared using CryoFilm (SECTION-LAB, Hiroshima, Japan) and a cryostat (CM-1850, Leica Microsystems, Wetzlar, Germany). The cryosections were washed with 50% ethanol once. In histochemical staining, the sections were stained with 30 μg/mL Nile red in 50% ethanol and 1 mg/mL Fluorescent Brightener 28 in 50% ethanol for 1 min each at room temperature of around 25 °C, then washed three times with 50% ethanol.

In immunostaining, the cryosections were washed with PBS three times, and the proteins in the sections were fixed with 4% paraformaldehyde in PBS for 20 min at room temperature, washed with PBS three times, followed by blocking with 5% BSA in PBS for 30 min and then washed three times. The sections were incubated with the primary antibody, anti-EuTPTs, diluted 1:100 in Can Get Signal® Immunostain Immunoreaction Enhancer (TOYOBO), for 12 h at 4 °C. After washing with PBS three times, the cryosections were incubated with Alexa Fluor 647-conjugated secondary antibody, diluted 1:100 in Can Get Signal® Immunostain Immunoreaction Enhancer, for 2 h at room temperature. The immunostained cryosections were further stained with Nile red and Fluorescent Brightener 28. The stained cryosections were placed on a glass slide, mounted in SlowFade® Gold (ThermoFisher Scientific, Yokohama, Japan), and then sealed with nail varnish.

### Microscopic analysis.
The spectral confocal laser scanning microscopy (SCLSM) analysis of the stained samples was performed at 25 °C using the SCLSM system (Digital Eclipse C1si; Nikon) equipped with CFI S Fluor 4×, CFI Plan Apo10×, 20×, 40×, and VC60×H lenses and the EZ-C1 3.40 software (Nikon). Fluorescence was excited with the 408-nm line of a blue diode laser, the 488-nm line of a solid laser, and the 640-nm line of a diode laser. The emission spectra in the ranges of 460–470 nm, 500–650 nm, and 640–750 nm with 5 nm bandwidths were recorded for detecting the cell wall, TPI and lipid, and Alexa Fluor 647, respectively, with TPI and lipid simultaneously detected in the rage of 500–650 nm. Reference samples were prepared by dissolving commercial *trans*-polyisoprene in chloroform at a final concentration of 1 mg/mL, and thin films were prepared on glass slides. The films were stained with Nile red, and fluorescence spectra from each of 50 locations (regions of interest) were measured and averaged. The fluorescence spectra of the stained samples were obtained from 10 to 25 locations and averaged. Images were acquired and averaged from five successive scans to improve the signal-to-noise ratio. Image processing, including spectral unmixing, was performed using EZ-C1 3.40 software and Adobe Photoshop CS4.

### Estimation of expression levels of EuTPTs.
Total RNA from *E. ulmoides* fruit was isolated as described above, and total RNA from transgenic tobacco plants was isolated using the Maxwell® 16 LEV Plant RNA Kit and Maxwell® 16 Automated Purification System (Promega, Madison, WI). Quantitative real-time PCR with the GoTaq qPCR Master Mix (Promega) was performed using the prepared cDNA preparations as templates for the StepOne Real-Time PCR (Applied Biosystems, Foster City, CA). The primer sets used in this analysis are listed in Supplementary Data 2. In every real-time PCR run, the *ACT* gene was used as a control to normalize the amount of cDNA template. The relative expression levels were calculated using the lowest *EuTPTs* transcript levels as the expression of 1.0.

### Measurement of TPI content.
The area of each *E. ulmoides* fruit was measured using Image J. The *E. ulmoides* seed was removed from the fruit. Then, the seed and the residual pericarp were completely dried at 55 °C for 2 days, and their weights were measured. Dried pericarp was ground into a fine powder using CryoMil MM400 (Retsch, Hann, Germany) for 1 min at 30 rpm with cooling using liquid nitrogen. The powder was suspended in ethanol, followed by centrifugation at 20,000 × g for 1 min, and the supernatant was discarded. This step was repeated until the supernatant was colorless. Five mg of the pericarp powder was dissolved in 1 mL chloroform with polybutadiene rubber dissolved at a concentration of 1 mg/mL as an internal standard and mixed gently for 20 h. In all, 5 μL of supernatant was analyzed by pyrolysis gas chromatography/mass spectrometry (PyGC/MS)[46]. PyGC/MS analysis was performed with ISQ7000, TRACE 1310 (Thermo Fisher Scientific), and Multi-Shot Pyrolyzer EGA/PY-3030D (FrontierLab, Fukushima) using a Ultra ALLOY® Capillary Column (30 m × 0.25 mm, 0.25-μm-thick film; ThermoFisher Scientific). TPI was pyrolyzed at 600 °C. The temperatures of the injector, GC/MS transfer line, and ion source were 200 °C, 250 °C, and 200 °C, respectively. The column temperature program was set at 5 min at 40 °C, and then increased by 8 °C/ min up to 300 °C, and kept at 300 °C for 5 min. The split ratio was set to 100:1. The mass range was set from 29 to 550 *m/z*.

### Molecular weight analysis of TPI.
The fine powder of pericarp was dissolved in toluene and heated at 60 °C with gently agitating for 6 h. The toluene extracted product was centrifuged at 1000 × g for 5 min, and the supernatant was collected and dried using an evaporator.

Two mg of the TPI was dissolved in 2 mL of tetrahydrofuran (THF) and gently mixed at room temperature for 2 days. The sample was filtrated using a 0.45-μm

membrane before size exclusion chromatography analysis. The size exclusion chromatography was performed with a Tosoh HLC-8320GPC EcoSEC chromatograph (Tosoh, Tokyo, Japan) using a guard column, TSKgel GMHhr-H (S) (5 μm particle size; I.D. × L, 7.5 mm × 7.5 cm; Tosoh), and two TSKgel GMHhr-H (S) (5 μm particle size; I.D. × L, 7.8 mm × 30 cm; Tosoh) columns with a molecular weight exclusion limit of 400,000,000. THF was used as the mobile phase at a flow rate of 0.5 mL/min at 40 °C. TPI was monitored by a Viscotek 270 Dual Light Scattering Detector (Malvern Instruments, Worcestershire, UK). The system was calibrated with the PolyCAL™ PS standard (Mw: 235,702; Mn: 111,713) (Malvern).

### Growth curve estimation of *E. ulmoides* fruit.
The 15 reproductive organs of *E. ulmoides* were sampled weekly from March to September in 2013 and 2015. The fresh samples were scanned using the CanoScan LiDE 220 (Canon Inc, Tokyo, Japan), and the area of each sample was calculated by ImageJ[47]. Because the dry weight of fruit, pericarp and seed, fruit area and TPI content might converge to the equilibrium points, the growth curves of each of these phenotypes were fitted with a logistic equation. The parameter estimations for the five logistic equations were conducted using the least squares method, and the accuracy of the estimated equations was confirmed by the $r^2$ value for the comparison between the estimated and measured scores. The analyses were conducted using R ver. 3.2.3.

### Isolation and analysis of washed TPI protein.
Fresh *E. ulmoides* pericarps stored at −80 °C were ground into a fine powder using CryoMil MM400 for 3 min at 30 rpm with cooling using liquid nitrogen. The fine powder was suspended in the equal volume of ice-cold 100 mM phosphate buffer pH 7.5, 5 mM MgSO₄, 5 mM 2-mercaptoethanol, Complete EDTA-free protease inhibitor cocktail (Roche, Diagnostics, Mannheim, Germany), and 0.9 g/mL polyvinylpyrrolidone, followed by vigorous vortexing and centrifugation at 2400 × g for 10 min. The unwashed, creamy, and floating fraction was transferred to a new tube, washed with 100 mM phosphate buffer pH 7.5, 5 mM MgSO₄, and 10 mM dithiothreitol, and centrifuged at 2400 × g for 10 min. Washed TPI was resuspended in 100 mM phosphate buffer, pH 7.5, 5 mM MgSO₄, 10 mM dithiothreitol, 0.5% CAHPS, and washed TPI protein was solubilized on ice by sonicating ten times for 30 s. Then, the sample was centrifuged at 20,000 × g for 15 min at 4 °C, and the aqueous phase was filtered using a 0.22-μm filter to remove remaining traces of TPI from the washed TPI protein solution. The TPI pellet was dried completely, and the weight was measured. Depending on the total weight of TPI, the washed TPI protein solution (the volume equivalent of 1 to 7.5 mg TPI) was desalted by methanol/chloroform extraction and acetone precipitation and dissolved in sample buffer for SDS-PAGE.

Purified TPI proteins were separated using SuperSep™ Ace 10–20% gradient gels (Wako, Osaka, Japan) and analyzed by silver staining or western blotting with the anti-EuTPT5 antibody.

### X-ray structural analysis.
Purified EuTPT3 and EuFPS1 were concentrated to 10 mg/mL in a buffer containing 20 mM Tris-HCl, pH 7.5, 150 mM NaCl, and 1 mM DTT. Prior to crystallization, aliquots of two more buffer components were added to the EuTPT3 preparation at the following final concentrations, 10 mM risedronate and 5 mM MgCl₂, for a 20-min incubation on ice. Crystals were obtained by hanging drop vapor diffusion at 20 °C (1.0 μL protein + 0.3 μL reservoir solution). EuTPT3 crystals (form1 and form2) were obtained in a reservoir solution of 9% PEG 3350 and 0.1 M dl-malic acid. To crystallize EuTPT3 (C94Y/A95F) mutant, a truncated version of EuTPT3 (residues 66–70 of EuFPS1 were replaced with GG) was used. The crystal of EuTPT3 (C94Y/A95F) was obtained in a reservoir solution of 0.1 M CHES, pH 9.5, and 20% PEG 8000. EuFPS1 crystal was obtained in a reservoir solution of 0.1 M HEPES, pH 7.0, and 25% PEG1000, 8% ethylene glycol, and 1 mM MgCl₂. Crystals were cryoprotected by the addition of ~30% ethylene glycol and flash-cooled in a 100-K nitrogen stream.

X-ray diffraction data were collected at 1.0 or 0.9 Å wavelength on the SPring-8 beamline 26B1 and 44XU (Hyogo, Japan). Diffraction data sets were indexed, integrated, and scaled using XDS. The EuTPT3 and EuFPS1 structures were determined by molecular replacement using Molrep with a search model of *Artemisia spiciformis* FPP synthase (Chain A from PDB ID 4KK2). The EuTPT3 (C94Y/A95F) structure was determined by molecular replacement using Molrep with a search model of the EuFPS1 dimer. Several rounds of refinement using phenix.refine or REFMAC5 and manual model building with Coot were performed. Stereochemical parameters were checked using the Molprobity suite in PHENIX. The final coordinates of EuTPT3 WT form 1, EuTPT3 WT form 2, EuTPT(C94Y/A95F), and EuFPS1 have been deposited in protein data bank with accession codes 7BUU, 7BUV, 7BUW and 7BUX, respectively. Structures contained 0.90%, 1.45%, 1.31%, and 0.15% Ramachandran outliers, respectively. Clash scores of the structures were 12.81, 11.23, 6.56, and 9.01, respectively. The figures were prepared using PyMol (http://www.pymol.org).

### Statistics and reproducibility.
All statistical analyses were performed using Microsoft Office Excel. Data are expressed as means ± standard deviation. All the experiments were evaluated at least three biological replicates with similar results.

**Reporting summary**. Further information on research design is available in the Nature Research Reporting Summary linked to this article.

## Data availability

All data generated or analyzed during this study are included in this published article (and its supplementary information files). The coordinates and structure factors for the crystal structures reported in this paper were deposited in the Protein Data Bank with accession codes 7BUU, 7BUV, 7BUW and 7BUX.

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

## Acknowledgements

We thank Dr. Hirotaka Uefuji for his technical support and discussion. This work was supported by Grants-in-Aid for Scientific Research [grant numbers 16H00783, 17H05732, 18K06094, 19H04735, and 19K07582 to H.M.] from the Japan Society for the Promotion of Science (JSPS), Yamada Science Foundation Grants-in-Aid, Eno Science Foundation Grants-in-Aid, and the New Energy and Industrial Technology Development Organization (NEDO) Japan. This work has been performed under the approval of the Photon Factory and SPring-8 Program Advisory Committee (Proposal Nos. 2017A6748, 2017A2570, 2017B6748, 2018A2719, 2018A6859, and 2017G702).

## Author contributions

H.K., N.S., and H.M. designed the research. H.K. and Y.K. performed tetrad analyses of *S. cerevisiae*. H.K., S. Takeno, and K.J.T. established the analytical method of PyGC/MS. T.Yoshizawa, T. Yamashita, and H.M. performed X-ray structural analysis and constructed homology models of the EuTPTs and H.K., T. Yoshizawa., S. Tanaka, and H.M. performed the structural comparisons. H.K. and Y.T. performed all the statistics analysis. H.K. performed all the other experiments. H.K., T. Yoshizawa, Y.T. N.S. and H.M. wrote the manuscript. Y.K., K.F., and Y.N. supervised the research.

## Competing interests

The authors declare no competing interests.
