## [Peer Review File · Communications Biology]

Reviewers' comments:

Reviewer #1 (Remarks to the Author):

Sequential isopentenyl diphosphate condensation by trans-prenyltransferase primarily produces short-chain prenyl precursors. Some plant trans-1,4-prenyltransferases (TPTs) produce ultrahigh molecular weight trans-1,4-polyisoprene (TPI), but its function is unclear. In this study, three novel functional ultralong-chain TPTs were identified and the potential molecular mechanism of this biosynthetic pathway was investigated based on structural analyses that differ substantially from FPS. Furthermore, the physiological function of EuTPT5 was also elucidated. These results provide new information on the mechanisms of catalysis and product chain elongation. Overall, it is a topic of interest to researchers in related areas and I recommend publishing it after minor revision. My detailed comments are as follows:

1. EuTPT1, 3, and 5 are novel Golgi-localized prenyltransferases, whereas FPSs reside in the cytosol or mitochondria. So, I am just wondering that if there are any correlation between the location of TPTs and ultrahigh molecular weight trans-1,4-polyisoprene biosynthesis?
2. The authors mentioned that EuTPT3 and EuTPT5 utilized all allyl substrates as reaction primers, whereas EuTPT1 used GPP, FPP, and GGPP as an allyl substrate, but not DMAPP. However, EuTPT1 had much higher activity than EuTPT3 and EuTPT5. What are the possible reasons for the apparent difference? Can the authors explain more?
3. Considering the difference of substrate spectrum and enzyme activity, I suggested that the enzymatic properties of these proteins should be evaluated.
4. In the section of "Structural comparisons of EuTPT3 and EuFPS1" (page 10 to 11), the authors concluded that the replacement of only two residues switched the dimeric conformation from the TPT-type to the FPS-type dimer. What if the residues, Tyr88 and Phe89 in EuFPS1 were mutated to Cys94 and Ala95, respectively? Could it result in the change of FPS-type dimer to EuTPT3 dimeric conformation? Could ultrahigh molecular weight TPI be biosynthesized by mutating these residues or rearranging $\alpha 5$ in FPS? Please comment.
5. I wonder if TPI could be biosynthesized in engineered *Escherichia coli* by the introduction of MVA (or MEP) pathway and EuTPT3.

Reviewer #2 (Remarks to the Author):

The manuscript entitled "Functional and mechanistic insights into ultrahigh molecular weight trans-1,4-polyisoprene biosynthesis by novel plant prenyltransferases" by Kajiura et al. describes the identification and characterization of trans-type prenyltransferases from *Eucommia ulmoides*. This study should attract much attention to the isoprene community as many remain unclear regarding the biosynthesis of ultrahigh molecular long-chain, either composed via cis- or trans-configuration. The authors conducted a series of experiments to detail the catalytic activity, three-dimensional structure, and in vivo function/localization of all or some of the three TPTs. While the novelty of this study assured, some main questions should be addressed before it is published.

1. The in vitro activity measurement indicates that EuTPT3 shows the lowest activity amongst all three TPTs. It seems that it also has distinct substrate preference of using GGPP as a precursor (FPP for the other two). It is understandable that not every protein can be crystallized or diffracted. But can the authors discuss the catalytic behaviors of TPT3 through structural point of view (plus sequence alignment of all three TPTs perhaps?).
2. The crystal structure of EuTPT3 was solved but the quality of crystal structure is considered rather low. The B values and Rwork/Rfree ratios are a bit high. Can the authors provide the Clashscore as

well? The resolution is low for TPT3 datasets so maybe additional parameters are required to judge the data quality.

3. Only the structural and functional analyses of TPT3 are available but the authors eventually found that TPT5 is the main long-chain producer in vivo. Can the authors at least perform functional analyses on TPT5 as well? Such as product measurement of equivalent CA-to-YF double mutant of TPT5. What are the protein sequence identity among TPTs?

Minor:

The term "ultrahigh molecular weight isoprene" should be elaborately defined. The descriptions of the carbon number (or length) of UPP, dolichol and natural rubber should be helpful.

Line 172-182, the residues constructing the larger tunnel of EuTPT3 should be displayed on the graphs.

The notions for structural elements should be unified throughout the MS. $\alpha 1-3$ or $\alpha 1-\alpha 3$?

The wavelength is 0.90000 to diffract TPT3 form 2 crystal and 1.00000 for the others. And only C94Y/A95F was collected in BL26B1 while others were in BL44XU. Is this correct?

Point-by-point responses to the previous review

First, we would like to thank the reviewer for his/her invaluable comments and suggestion to improve our manuscript. In the revised manuscript, we corrected the words the reviewers pointed out, described the points the reviewers brought up, and the changed parts *in Red*.

Reviewer 1

Sequential isopentenyl diphosphate condensation by trans-prenyltransferase primarily produces short-chain prenyl precursors. Some plant trans-1,4-prenyltransferases (TPTs) produce ultrahigh molecular weight trans-1,4-polyisoprene (TPI), but its function is unclear. In this study, three novel functional ultralong-chain TPTs were identified and the potential molecular mechanism of this biosynthetic pathway was investigated based on structural analyses that differ substantially from FPS. Furthermore, the physiological function of EuTPT5 was also elucidated. These results provide new information on the mechanisms of catalysis and product chain elongation. Overall, it is a topic of interest to researchers in related areas.

Thank you for taking the time to review and give us your kind comments and constructive suggestions. We are honor to have our manuscript evaluated well.

1. EuTPT1, 3, and 5 are novel Golgi-localized prenyltransferases, whereas FPSs reside in the cytosol or mitochondria. So, I am just wondering that if there are any correlation between the location of TPTs and ultrahigh molecular weight trans-1,4-polyisoprene biosynthesis?

Thank you for your very interesting question. IPP and FPP, substrates for TPI biosynthesis, are biosynthesized via mevalonate pathway in the cytosol with their hydrophilicities and/or solubility to cytosolic fluid. These IPP and FPP are essential for biosynthesis of secondary metabolites and phytohormones used in a various organelle and thus it is rather convenient not to be biosynthesized with a specific organelle for cell homeostasis and viability. On the other hand, TPI is hydrophobic molecule and could not present itself in cytosol, and therefore it should be first synthesized in between the lipid layered some organelles, such as the ER, nucleus, mitochondria, chlorophyll, or Golgi. This speculation is supported by the biosynthetic and accumulation mechanisms of natural rubber which is first

synthesized in the ER bilayer with using IPP substrate supplied from cytosol. We consider TPI is also biosynthesized with same manner as natural rubber biosynthesis but occurred in Golgi. We speculate that the difference of localization between TPTs and FPS are presumably from the effect of some amino acid insertions in TPTs as shown in supplemental Fig. 1 and described in the main text.

2. The authors mentioned that EuTPT3 and EuTPT5 utilized all allyl substrates as reaction primers, whereas EuTPT1 used GPP, FPP, and GGPP as an allyl substrate, but not DMAPP. However, EuTPT1 had much higher activity than EuTPT3 and EuTPT5. What are the possible reasons for the apparent difference? Can the authors explain more?

Thank you for your insightful comment. Previously we reported the possibility of competitive effect of IPP and DMAPP for FPP biosynthesis. All *trans*-prenyltransferase including FPS possesses have two DDxxD motifs essential for substrate binding and activity. Interestingly, high concentration of IPP and DMAPP inhibit FPS activity presumably due to the possibility that IPP and DMAPP are structurally similar and could bind to both DDxxD motif. However, GPP did not have any inhibitory effect on FPS activity. EuTPT1 has the strongest activity among three TPTs, and therefore turnover k_{cat} of EuTPT1 is also considered to be higher than other two EuTPTs, leading to “rough recognition” of substrate. In other word, instead of IPP, DMAPP might also bind to second DDxxD motif which site is IPP binding site, resulting in showing in inhibitory effect on EuTPT1 activity. This speculation is supported by the results of EuFPSs which shows that the higher activity of FPS exhibits the more roughly end product synthesis. Therefore, we consider it is reasonable that weaker activity recognizes the substrate with higher selectivity and produces the final product.

In terms of comparisons among EuTPTs from structural point of view, we added Supplementary Fig. S15-16 and inserted the following sentences

Discussion, page 15, line 3;

Among the TPI synthases, EuTPT3 shares 77% and 72% amino acid identity with EuTPT1 and EuTPT5 (Supplementary Fig. S15). Although the most residues around the DDxxD motifs are structurally conserved, EuTPT3 has

lower activity than EuTPT1 (Fig. 1c). Residues Ile164-Ala165 on $\alpha 6$ in EuTPT3 are occupied by Val164-Ala165 in EuTPT1 and Val165-Ala166 in EuTPT5, respectively (Supplementary Fig. S15). Crystal structure of EuTPT3 showed that the residues Ile164-Ala165 contact with Ile164 and Ile125 of the neighboring subunit in the hydrophobic tunnel, suggesting that smaller side chains such as valine and glycine provide the wider path for the product that may contribute to the activity difference (Fig. 4c and Supplementary Fig. S16)".

3. *Considering the difference of substrate spectrum and enzyme activity, I suggested that the enzymatic properties of these proteins should be evaluated.*

Thank you for your suggestion. We added the basic properties of EuTPTs, such as optimal temperatures, optimal pHs, and ion dependencies as a supplemental data (Supplemental Table S1) and the following sentence;

page 6, line 21;

All EuTPTs showed their activities around neutral pH and mild temperature of 20-40°C and required divalent cation (Supplemental Table S1).

4. *In the section of "Structural comparisons of EuTPT3 and EuFPS1" (page 10 to 11), the authors concluded that the replacement of only two residues switched the dimeric conformation from the TPT-type to the FPS-type dimer. What if the residues, Tyr88 and Phe89 in EuFPS1 were mutated to Cys94 and Ala95, respectively? Could it result in the change of FPS-type dimer to EuTPT3 dimeric conformation? Could ultrahigh molecular weight TPI be biosynthesized by mutating these residues or rearranging $\alpha 5$ in FPS? Please comment.*

Thank you for your critical question. In the revised manuscript, we did not determine the crystal structures, but performed the mutational analysis of all EuTPTs and EuFPSs to examine the effect of the presence of bulky amino acid residues on the end product preferences. As you expected, the mutations to bulky amino acids resulted in the synthesis of short chain isoprene, such as FPP and GGPP, whereas the mutations to amino acid which has smaller side chain led to the synthesis of end product larger chain length than GGPP. However, the mutations to amino acid with smaller side chains produced much shorter than the

ultrahigh molecular weight TPI, indicating that EuTPTs have other critical amino acid(s) to biosynthesize ultrahigh molecular weight TPI. In other words, mutations of Tyr88 and Phe89 in EuFPS1 presumably are not enough to transform the dimer conformation from FPS-type to TPT-type. We added this result as a supplemental data (Supplemental Fig. S8) and also the following sentence;

page 10, line 7;

Actually, the replacements of the exact two amino acids in other EuTPTs and their endo product analysis provided reasonable results; mutations of Cys94 and Ala95 in EuFPS1 and Cys96 and Ala96 in EuTPT5 to Tyr and Phe, respectively, resulted in synthesizing FPP, whereas Tyr88 and Phe89 in EuTPT2 (EuFPS1) and Phe95 and Phe96 in EuTPT4 (EuFPS2) produced longer chain length end product than FPP (Supplementary Fig. 8).

5. I wonder if TPI could be biosynthesized in engineered Escherichia coli by the introduction of MVA (or MEP) pathway and EuTPT3.

Thank you for your very interesting question. We consider that unfortunately, ultrahigh molecular weight TPI could not be synthesized even in the engineered bacteria due to the solubility of TPI and the lack of specific organelle to accumulate TPI. The engineered bacteria might produce much smaller amount of ultrahigh molecular weight TPI, but its small granules in the cell stresses and/or damages the cell, resulting in showing the lethality. In the natural rubber biosynthesis study, they use wheat germ system for in vitro synthesis of ultrahigh molecular weight CPI. We speculate

Reviewer 2

*The manuscript entitled “Functional and mechanistic insights into ultrahigh molecular weight trans-1,4-polyisoprene biosynthesis by novel plant prenyltransferases” by Kajiura et al. describes the identification and characterization of trans-type prenyltransferases from *Eucommia ulmoides*. This study should attract much attention to the isoprene community as many remain unclear regarding the biosynthesis of ultrahigh molecular long-chain, either composed via cis- or trans-configuration. The authors conducted a series of experiments to detail the catalytic activity, three-dimensional structure, and in vivo function/localization of all or some of the three TPTs. While the novelty of this study assured, some main questions should be addressed before it is published.*

1. The in vitro activity measurement indicates that EuTPT3 shows the lowest activity amongst all three TPTs. It seems that it also has distinct substrate preference of using GGPP as a precursor (FPP for the other two). It is understandable that not every protein can be crystallized or diffracted. But can be the authors discuss the catalytic behaviors of TPT3 through structural point of view (plus sequence alignment of all three TPTs perhaps?).

Thank you for your insightful comment. In terms of comparisons among EuTPTs from structural point of view, we added Supplementary Fig. S15-16 and inserted the following sentences

Discussion, page 15, line 3;

Among the TPI synthases, EuTPT3 shares 77% and 72% amino acid identity with EuTPT1 and EuTPT5 (Supplementary Fig. S15). Although the most residues around the DDxxD motifs are structurally conserved, EuTPT3 has lower activity than EuTPT1 (Fig. 1c). Residues Ile164-Ala165 on $\alpha 6$ in EuTPT3 are occupied by Val164-Ala165 in EuTPT1 and Val165-Ala166 in EuTPT5, respectively (Supplementary Fig. S15). Crystal structure of EuTPT3 showed that the residues Ile164-Ala165 contact with Ile164 and Ile125 of the neighboring subunit in the hydrophobic tunnel, suggesting that smaller side chains such as valine and glycine provide the wider path for the product that may contribute to the activity difference (Fig. 4c and Supplementary Fig. S16)".

2. The crystal structure of EuTPT3 was solved but the quality of crystal structure is considered rather low. The B values and Rwork/Rfree ratios are a bit high. Can the authors provide the Clashscore as well? The resolution is low for TPT3 datasets so maybe additional parameters are required to judge the data quality.

Thank you for your valuable comment. We provided clash scores and molprobability scores in Supplementary Table S1.

3. Only the structural and functional analyses of TPT3 are available but the authors eventually found that TPT5 is the main long-chain producer in vivo. Can the authors at least perform functional analyses on TPT5 as well? Such as product measurement of equivalent CA-to-YF double mutant of TPT5. What are the protein sequence identity among TPTs?

Thank you for your question. As you suggested, not only EuTPT5, but we performed the mutational analysis of all EuTPTs and EuFPSs to examine the effect of the presence of bulky amino acid residues on the end product preferences. As you expected that CA-to-YF double mutant of TPT5 resulted in the synthesis of much shorter TPI, *i.e.* FPP. We added the results (Supplemental Fig. S8) and the sequence identity among (Supplemental Fig. S15) TPT as supplemental data.

Minor:

The term “ultrahigh molecular weight isoprene” should be elaborately defined. The descriptions of the carbon number (or length) of UPP, dolichol and natural rubber should be helpful.

Thank you for your invaluable suggestion. We added the information of definition of chain length in Abstract and Introduction parts of the revised manuscript as follows;

Abstract,

Some plant *trans*-1,4-prenyltransferases (TPTs) produce ultrahigh molecular weight *trans*-1,4-polyisoprene (TPI) with a molecular weight of over 1.0 million,

Introduction;

However, some plant species, including *Eucommia ulmoides*, *Palaquium gutta*, *Manilkara bidentata*, and *Manilkara zapota*, produce ultrahigh molecular weight

TPI with a chain length of C_{100} or much more ($C_{50,000}$, estimated from the maximum molecular weight)

Line 172-182, the residues constructing the larger tunnel of EuTPT3 should be displayed on the graphs.

According to your suggestion, we made a figure (Fig. 4c) showing the residues constructing the larger tunnel of EuTPT3 and added following sentence;

page 8, line 18;

The tunnel is mainly composed by hydrophobic residues of $\alpha 5$ and $\alpha 6$ (Fig. 4c).

The notions for structural elements should be unified throughout the MS. $\alpha 1-3$ or $\alpha 1-\alpha 3$?

Thank you for your comment. We unified the structural elements as $\alpha 1-\alpha 3$ in the manuscript.

The wavelength is 0.90000 to diffract TPT3 form 2 crystal and 1.00000 for the others. And only C94Y/A95F was collected in BL26B1 while others were in BL44XU. Is this correct?

We have double checked the log files. The wavelengths and beamlines are correct as described in the manuscript.

REVIEWERS' COMMENTS:

Reviewer #1 (Remarks to the Author):

The authors have carefully revised the manuscript and addressed the comments raised by the reviewers. Considering the novelty of this study, it was suggested to be accepted for publication.

Reviewer #2 (Remarks to the Author):

The questions raised have been properly answered in the revised manuscript. The reviewer considers the manuscript now suitable to be published.